# UNSUPERVISED 3D SCENE REPRESENTATION LEARNING VIA MOVABLE OBJECT INFERENCE

## ABSTRACT

Unsupervised, category-agnostic, object-centric 3D representation learning for complex scenes remains an open problem in computer vision. While a few recent methods can now discover 3D object radiance fields from a single image without supervision, they are limited to simplistic scenes with objects of a single category, often with a uniform color. This is because they discover objects purely based on appearance cues—objects are made of pixels that look alike. In this work, we propose Movable Object Radiance Fields (MORF), aiming at scaling to complex scenes with diverse categories of objects. Inspired by cognitive science studies of object learning in babies, MORF learns 3D object representations via movable object inference. During training, MORF first obtains 2D masks of movable objects via a self-supervised movable object segmentation method; it then bridges the gap to 3D object representations via conditional neural rendering in multiple views. During testing, MORF can discover, reconstruct, and move unseen objects from novel categories, all from a single image. Experiments show that MORF extracts accurate object geometry and supports realistic object and scene reconstruction and editing, significantly outperforming the state-of-the-art.

## 1 INTRODUCTION

Learning object-centric 3D representations of complex scenes is a critical precursor to a wide range of application domains in vision, robotics, and graphics. The ability to factorize a scene into objects provides the flexibility of querying the properties of individual objects, which greatly facilitates downstream tasks such as visual reasoning, visual dynamics prediction, manipulation, and scene editing. Furthermore, we hypothesize that building factorized representations provides a strong inductive bias for compositional generalization (Greff et al., 2020), which in turn enables the model to understand novel scenes with previously unseen objects and configurations.

While supervised learning methods have shown promise in learning 3D object representations (such as neural radiance fields (Mildenhall et al., 2020)) from images (Ost et al., 2021; Kundu et al., 2022; Müller et al., 2022), they rely on annotations of specific object and scene categories. A recent line of work (Yu et al., 2022; Stelzner et al., 2021) has explored the problem of unsupervised discovery of object radiance fields. These models can be trained from multi-view RGB or RGB-D images to learn object-centric 3D scene decomposition without annotations of object segments and categories. However, they are only demonstrated to work well on simplistic scenes. It remains challenging to obtain accurate reconstructions on more complex datasets. A fundamental reason is that they heavily rely on visual appearance similarity to discover object entities, which limits their scalability beyond simple texture-less objects.

In this work, we aim to scale unsupervised 3D object-centric representation learning to complex visual scenes with textured objects from diverse categories. To this end, we propose a Movable Object Radiance Fields (MORF) model, which learns to infer 3D object radiance fields from a single image. Rather than appearance similarity, the underlying principle of MORF uses to discover object entities is material coherence under everyday physical actions, i.e., an object is movable as a whole in 3D space (Spelke, 1990). However, it is challenging to obtain learning signals to directly infer movable objects in 3D. MORF addresses this problem by integrating a recent self-supervised 2D movable object segmentation method, EISEN (Chen et al., 2022), to extract movable object segments in 2D images, as well as differentiable neural rendering to bridge the gap between 2D learning signals and

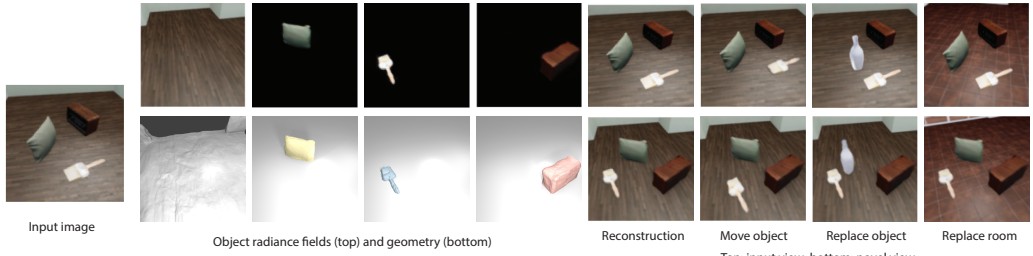

Input image

Object radiance fields (top) and geometry (bottom)

Reconstruction    Move object    Replace object    Replace room

Top: input view; bottom: novel view

Figure 1: Illustration of unsupervised, category-agnostic, object-centric 3D representation learning. Given a single image, our goal is to infer object radiance fields that allow photometric and geometric 3D reconstruction. This factorized representation enables 3D scene manipulation, including moving object and replacing the background.

**3D inference.** MORF learns to conditionally infer object radiance fields from the segmented images, which provide strong inductive bias for object-centric factorization of 3D scenes.

Specifically, we pretrain EISEN on optical flow from unlabeled videos. EISEN learns object image segmentations by perceptually grouping parts of a scene that would move as cohesive wholes, serving as a module that estimates high-quality object segmentations on static images. After pretraining, MORF learns to extract object-centric latent representations from segmented images and generate object radiance fields from the factorized latents. To facilitate high-quality reconstruction of textured objects, our latent object representation consists of both an entity-level latent and a pixel-level latent that better encodes appearance details.

To evaluate our method, we propose a challenging dataset with a diverse set of realistic-looking objects, going beyond simplistic scenes considered by most current unsupervised 3D object discovery methods (Yu et al., 2022; Stelzner et al., 2021; Sajjadi et al., 2022b). We demonstrate that MORF can learn high-quality 3D object-centric representations in complex visual scenes, allowing photometric and geometric reconstruction for these scenes from single views (Figure 1). Moreover, our learned representations enable 3D scene manipulation tasks such as moving, rotating, and replacing objects, and changing the background of complex scenes. Beyond systematic generalization to unseen spatial layouts and arrangements, we further show that MORF is able to generalize to unseen object categories and appearances while maintaining reasonable reconstruction and geometry estimation quality.

In summary, our contributions are three-fold. First, we propose scaling the learning of unsupervised, category-agnostic, object-centric 3D representation learning beyond simplistic scenes by discovering objects with coherent motion, in addition to visual appearance. Second, to instantiate our idea, we propose Movable Object Radiance Fields (MORF), which integrates 2D movable object segmentation with neural rendering to allow 3D movable object discovery. Third, we demonstrate that MORF allows photometric and geometric reconstruction and editing of complex 3D scenes with textured objects from diverse unseen categories.

## 2 RELATED WORK

**Unsupervised 2D object discovery** Our method is closely related to recent work on unsupervised scene decomposition, which aims to decompose multi-object scenes into separate object-centric representations from images without human annotations. Most works formulate the problem as learning compositional generative models in the 2D image space. They decompose a visual scene into a set of localized object-centric patches (Eslami et al., 2016; Crawford & Pineau, 2019; Kosiorek et al., 2018; Lin et al., 2020; Jiang et al., 2019a) or a set of scene mixture components (Burgess et al., 2019; Greff et al., 2019; 2016; 2017; Engelcke et al., 2019; Locatello et al., 2020; Monnier et al., 2021; Jiang et al., 2019b). The scene mixture models typically generate single-object RGBA images and blend them to reconstruct the full-scene images using iterative inference with recurrent networks (Burgess et al., 2019) or set-based convolutional encoders (Locatello et al., 2020). However, these methods have so far been unable to scale to complex real-world images. A recent branch of work on self-supervised object segmentations explores additional supervision signals such as motions and depth for learning object segmentations (Bear et al., 2020; Kipf et al., 2021; Chen et al., 2022;

Bao et al., 2022; Elsayed et al., 2022; Ye et al., 2022). However, these 2D methods are not aware of the 3D nature of scenes, and thus they do not provide 3D understanding of the underlying scenes.

**Unsupervised 3D object discovery.** Discovering objects from image collections has been a long-standing topic in computer vision, but earlier works on object discovery (a.k.a. co-segmentation) represent objects as 2D image segments without 3D information (Russell et al., 2006; Sivic et al., 2005; 2008; Grauman & Darrell, 2006; Joulin et al., 2010; Rubio et al., 2012; Vicente et al., 2011; Rubinstein et al., 2013; Cho et al., 2015; Li et al., 2019; Vo et al., 2020). Recently, some works have been focusing on discovering 3D object representations. A related branch of works focuses on 3D reconstruction from a single image (Ye et al., 2021; Kulkarni et al., 2019; 2020; Kanazawa et al., 2018; Wu et al., 2021). However, it requires strong category-specific shape priors, making it difficult to scale to complex real-world data. Elich et al. (2020) infer object shapes (Park et al., 2019) from a single scene image. Chen et al. (2020) extend Generative Query Network (Eslami et al., 2018) to decompose 3D scenes. Notably, the closest to our work is a recent branch of works that focuses on inferring 3D neural object representations from single images (Yu et al., 2022; Smith et al., 2022; Stelzner et al., 2021) or sparse views (Sajjadi et al., 2022b). However, these methods rely on visual appearance to discover object entities. This fundamental assumption makes them difficult to scale to complex scenes with textured objects, diverse object categories, or objects under different lighting. In contrast, our approach leverages motions as the underlying object concept, which is category-agnostic and generalizable to different object appearances.

**Scene de-rendering.** Aiming to provide full scene understanding, a line of scene de-rendering works have shown reconstructing 3D object-centric representations in specific types of scenes (Wu et al., 2017; Yao et al., 2018; Kundu et al., 2018; Ost et al., 2021; Kundu et al., 2022; Yang et al., 2021; Wu et al., 2022). Recently, Ost et al. (2021) propose to represent dynamic scenes into a scene graph where each node encodes object-centric information. Müller et al. (2022) recover 3D object information such as shape, appearance, and pose in autonomous driving scenes. However, these methods rely on manual annotations of object categories (such as cars) and scene categories (such as street scenes). Similarly, Gkioxari et al. (2022) propose a method that learns to predict 3D shape and layout for objects by relying on 2D bounding box supervision. Our approach only requires object motion in video for inferring object segmentations during training, without requiring manual annotations.

**Neural scene representations and rendering.** Our object representation is based on recent progresses in neural scene representations (Park et al., 2019; Mescheder et al., 2019; Sitzmann et al., 2019) and neural rendering (Tewari et al., 2020). Neural scene representations implicitly model 3D scenes using the parameters of deep networks, which could be learned from only 2D images (Niemeyer et al., 2020; Sitzmann et al., 2019) with differentiable rendering techniques (Kato et al., 2020; Tewari et al., 2020). Specifically, Neural Radiance Fields (NeRFs) (Mildenhall et al., 2020) has shown photorealistic scene modeling of static scenes using only images. The most relevant works in this line aim to infer NeRFs from a single image (Yu et al., 2020; Kosiorek et al., 2021; Rematas et al., 2021). While these works focus on single objects or holistic scenes, we address decomposing a multi-object scene without human supervision. Another relevant branch of works aims at incorporating NeRFs into compositional models (Niemeyer & Geiger, 2020; 2021), such as Niemeyer & Geiger (2020). While they target at scene synthesis, we instead focus on multi-object inference, which GIRAFFE cannot address (Yu et al., 2022).

## 3 MOVABLE OBJECT RADIANCE FIELDS (MORF)

We now describe the problem formulation and our approach, Movable Object Radiance Fields. Given a single input image of a scene that potentially has objects from diverse categories, our goal is to factorize the scene into a set of object-centric conditional radiance field representations. To allow unsupervised decomposition of such complex 3D scenes, we propose learning to discover movable objects by integrating 2D movable object inference with 3D-to-2D neural rendering.

Movable Object Radiance Fields therefore has three stages. First, it decomposes the input image by inferring its 2D object segmentation masks (Figure 2a). The segmentation extraction network is trained separately and self-supervised by optical flow for grouping scene elements that often move together into an object segment. Second, for each mask, Movable Object Radiance Fields learns an object radiance field with object and pixel latent codes for object features and locally varying details, respectively (Figure 2b). Finally, these object radiance fields are then composed to re-render the

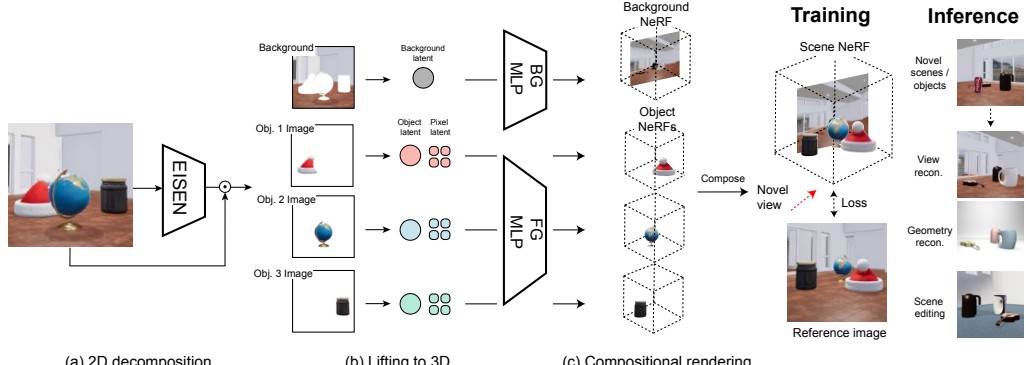

Figure 2: Illustration of Movable Object Radiance Fields (MORF). MORF takes as input a single image of a scene with potentially diverse objects, and infers 3D object and background radiance fields. (a) MORF integrates an image-based movable object inference method, EISEN, that predicts a set of object masks, which are used to create masked images as inputs for learning object radiance fields; (b) MORF generates object radiance fields conditioned on the latent object and pixel codes. (c) During training, MORF reconstructs the novel view via compositional rendering and is supervised by reconstruction losses. During inference, MORF takes a single view of a new scene, and infers object and background radiance fields in a single forward pass.

scene from multiple views, supervised by the reconstruction loss (Figure 2c). We now describe each component in detail.

## 3.1 MOVABLE OBJECT INFERENCE IN 2D

Given a single RGB image, we first compute a 2D object segmentation mask, represented as a $H \times W \times K$ binary tensor $\mathbf{M}^o$, where $K$ is the number of object masks. The background mask $\mathbf{M}^b$ can be computed by taking the complement of the object masks union: $\mathbf{M}^b = (\bigcup_{i=0}^{K} \mathbf{M}_i^o)^c$.

We adopt the EISEN architecture from Chen et al. (2022) for generating high-quality object segmentation masks. The core idea of EISEN is to construct a high-dimensional plateau map representation (of shape $H' \times W' \times Q$) for each image, in which all the feature vectors $q_{ij}$ belonging to the same object are aligned (i.e., have cosine similarity $\approx 1$) and all feature vectors that belong to other objects are nearly orthogonal (cosine similarity $\approx 0$). Given this representation, the object segments can be easily extracted from a plateau map by finding clusters of vectors pointing in similar directions.

More specifically, EISEN first applies a convolutional backbone on the input image to obtain a feature grid, followed by an affinity prediction module that computes pairwise affinities between features at pixel location $(i, j)$ and the features at its neighboring locations within a local window. Then EISEN constructs a graph, with nodes represented by $Q$-dimensional feature vectors and edges represented by the pairwise affinities. A message-passing graph neural network is run on the graph to construct the plateau map representations, by passing excitatory messages that align the feature vectors of nodes belonging to the same object and inhibitory messages that orthogonalize the feature vectors of nodes belonging to distinct objects. Once the plateau map representation is obtained, EISEN imposes winner-take-all dynamics on the plateau map to extract object segments. We refer the readers to Chen et al. (2022) for more implementation details of EISEN.

During training, EISEN learns the pairwise affinities from optical flow estimates from a RAFT (Teed & Deng, 2020) network pretrained on Sintel (Mayer et al., 2016). Consider a pair of scene elements $(a, b)$ that project into image coordinates $(i, j)$ and $(i', j')$ respectively. If only one is moving, it is usually the case that they do not belong to the same object; when neither is moving, there is no information about their connectivity. We use this physical logic to construct a pairwise supervision signal for EISEN's affinity matrix. The EISEN training loss is the masked row-wise KL divergence between the predicted and target connectivity. Although EISEN requires a frame pair as input for computing optical flow during training, it only requires a single static image during inference time for computing the object segmentation masks, and subsequently predicting the object radiance fields.

## 3.2 LEARNING OBJECT RADIANCE FIELDS IN 3D

We model the 3D representations of objects and backgrounds as conditional neural radiance fields. Given an input image and its predicted EISEN masks, we compute both global slot-based conditioning $\mathbf{s} \in \mathbb{R}^d_s$ and local pixel conditioning $\mathbf{p} \in \mathbb{R}^d_p$ for high-fidelity reconstruction of complex scenes with locally varying appearances. Since the geometry of the background and foreground objects are highly different, representing them using the same conditional NeRF might impede the system's capacity to model complex and diverse object geometries. Therefore, we parameterize the background NeRFs and object NeRFs using two separate conditional MLPs with parameters $\theta_b$ and $\theta_o$, respectively. The MLP parameters of the object NeRFs are shared across all the foreground objects. The latent codes are used as input to the MLP networks, along with 3D position encoding $\gamma(\mathbf{x})$ and view directions $d$.

$$f_{\theta_b}(\gamma(\mathbf{x}), \mathbf{d}|\mathbf{s}^b, \mathbf{p}^b) = (\sigma_b, \mathbf{c}_b), \tag{1}$$

$$f_{\theta_o}(\gamma(\mathbf{x}), \mathbf{d}|\mathbf{s}^i, \mathbf{p}^i) = (\sigma_i, \mathbf{c}_i), \quad i \in \{0, ..., K\} \tag{2}$$

To compute object latent codes, we adapt the slot-based update mechanism proposed by Locatello et al. (2020) for inferring object latents. Unlike the original formulation of uORF, which computes object slots using an attention module, we directly compute the initial object latents via average pooling of a convolutional feature map using the predicted EISEN segmentation masks $s^b = W^{bT} \cdot v^b(feat)$, where $W^b_i = \mathbf{M}^b_i/(\sum_{j=0}^N \mathbf{M}^b_i)$. The latent codes are iteratively updated via a learnable Gated Recurrent Unit (Chung et al., 2014), $s^b \leftarrow \text{GRU}^b(s^b, \text{updates}^b)$, producing the final latents for conditional neural radiance field. The latent codes $s^i$ for the foreground objects are computed using the same formulation.

To obtain the pixel latent codes, we first mask the input image using $\mathbf{M}^o$ and $\mathbf{M}^b$, and extract a convolution feature grid for the background and each object $i$ respectively $\mathbf{W}^b = E(\mathbf{M}^b(\mathbf{I}))$, $\mathbf{W}^o_i = E(\mathbf{M}^o_i(\mathbf{I}))$. For each query point $\mathbf{x}$ on a camera ray, we follow Yu et al. (2020) to retrieve the image features of each object by projecting $\mathbf{x}$ onto the image plane coordinates $\pi(\mathbf{x})$ and extract latent codes from the feature grid via bilinear interpolation, obtaining pixel latent codes $p^b = \mathbf{W}^b(\pi(\mathbf{x}))$ and $p^i = \mathbf{W}^b_i(\pi(\mathbf{x}))$.

## 3.3 COMPOSITIONAL RENDERING

At each point in the rendering of a scene, the final pixel value is a combination of contributions from each individual scene element to the 3D volumes projecting to that point. We follow uORF (Yu et al., 2022) and take a weighted average of the individual components to obtain the combined density $\boldsymbol{\sigma}$ and color $\mathbf{c}$. The composite volumetric radiance field is then rendered into a 2D image via the numerical integration of volume rendering by sampling $S$ discrete points along each pixel ray parameterized as $r(t) = o + td$, with the ray origin $o$ and ray unit direction vector $d$. The points on each ray are sampled between pre-specified depth bounds $[t_n, t_f]$, with distance $\delta_j$ between adjacent samples along the ray. Thus, the final pixel color is given by:

$$C(r) = \sum_{i=0}^S T_i[1 - \exp(-\boldsymbol{\sigma}_i\delta_i)]\mathbf{c}_i, \quad T_i = \exp(-\sum_{j=0}^{i-1}\boldsymbol{\sigma}_i\delta_i), \tag{3}$$

where

$$\boldsymbol{\sigma} = \sum_j p_j\sigma_j, \quad \mathbf{c} = \sum_j p_jc_j, \quad p_j = \frac{\sigma_j}{\sum_k \sigma_k}, \quad j, k \in \{b, 0, ..., K\}. \tag{4}$$

## 3.4 LOSS FUNCTION

During training, we randomly select a single image of a scene as input, and render multiple novel views. We train the model using both reconstruction loss and perceptual loss: $\mathcal{L} = \mathcal{L}_r + \lambda_p\mathcal{L}_p$ with $\lambda_p = 0.006$. The reconstruction loss is the L2 loss between the rendered image and the ground-truth image $\mathcal{L}_r = ||I - \hat{I}||^2$. Since reconstruction loss is sensitive to small geometric imperfections and often results in blurry reconstructions, especially due to uncertainties in rendering a novel view, we add a perceptual loss term to mitigate this problem. We compute the perceptual loss as $\mathcal{L}_p = ||e_k(I) - e_k(\hat{I})||^2$, where $e_k(\cdot)$ is k-th layer of an off-the-shelf VGG16 (Simonyan & Zisserman, 2014) image encoder $e$ with frozen pre-trained weights.

## 4 EXPERIMENTS

We demonstrate our approach on three tasks: (a) novel view synthesis, (b) scene geometry reconstruction, and (c) editing scenes by moving objects, replacing objects, and changing the background.

**Datasets** We generated three variants of a complex synthetic scene dataset using the ThreeDWorld simulation environment (Gan et al., 2020). Each scene includes four camera views with a random azimuth angle and a fixed elevation; the camera always points at the scene center.

*Playroom* dataset contains a wide range of realistically simulated and rendered objects. Each scene includes 3 objects randomly sampled from a set of 2000 object models. These models are drawn from a wide set of categories and have a range of complex 3D shapes and textures. They are placed in random positions and poses in rooms selected randomly from a collection of indoor environments with varying 3D room layouts and floor/wall textures. In each scene, one object is invisibly pushed to generate object motion. There are 15,000 scenes for training and 600 for testing.

*Playroom-novel* dataset contains novel object models for evaluating the generalization performance of the models. Each scene contains 3 objects randomly sampled from 100 distinct object models that are held out from the *Playroom* scenes. They are placed in the same room environments seen in the *Playroom* dataset. There are 600 scenes for evaluation.

*Playroom-edit* dataset is designed for evaluating a model's ability to manipulate object radiance fields and synthesize novel images. The dataset contains scenes that result from three types of editing: moving objects, replacing objects, and changing the background. For object moving, we randomly pick one object in the scene and teleport it to a random position. For object replacement, we switch a randomly selected object with an object from a different scene. For background replacement, we similarly switch the background with that of another scene. For each scene editing task, we render 200 test scenes for evaluation.

**Baselines** We compare MORF to the slot-conditioned unsupervised object radiance field method uORF (Yu et al., 2022) and nonfactorized pixel-conditioned method pixelNeRF (Yu et al., 2020). Both methods learn radiance fields from RGB images without ground-truth object annotations or depth supervision. We also compare to an ablated version of MORF trained without pixel latents to illustrate the benefits of local conditioning in 3D representation learning. We adopt the same training procedures and hyperparameter choices as reported in the original papers. For a fair comparison, all models receive input images with the same resolution and are trained with the same batch size.

### 4.1 NOVEL VIEW SYNTHESIS

We randomly select one camera view of each scene as input and use the remaining three images as ground truth for evaluating the quality of novel view synthesis. Besides uORF and pixelNeRF, we also compare with the pixel feature-ablated version of MORF. All the models are evaluated using the standard image quality metrics PNSR, SSIM (Wang et al., 2004), and LPIPS (Zhang et al., 2018).

**Results** As shown in Table 1 and Figure 3, MORF outperforms the baseline methods both quantitatively and qualitatively. uORF and SRT(Sajjadi et al., 2022a) are able to learn rough object decompositions and position estimates, but fail to represent the object shapes accurately, resulting in inaccurate rendering from novel views. The version of MORF without pixel features as conditioning performs second-best. Although both methods use object latent conditioning, the latter attains substantially better reconstructions, suggesting accurate object segmentations help constrain the optimization of the neural implicit function. Both MORF and pixelNeRF are able to render novel views reasonably well. We highlight that pixelNeRF's reconstructions are blurry both in background regions (such as floor tiles) and on objects (such as the dumbbell), while MORF's reconstructions are sharper. This demonstrates the importance of accurate object decomposition on high-quality novel view synthesis. The comparison to the pixel feature-ablated MORF illustrates the advantage of using pixel features over object latents in capturing fine-grained details of scenes.

### 4.2 GEOMETRY RECONSTRUCTION

To evaluate the quality of 3D scene representations, we first extract meshes from each model's learned density field with the marching cubes algorithm (Lewiner et al., 2003). We compute the density field by evaluating the foreground decoder at grid points in the world coordinate system. For a fair comparison, all the models are evaluated using the same grid size of 256.

*Playroom*

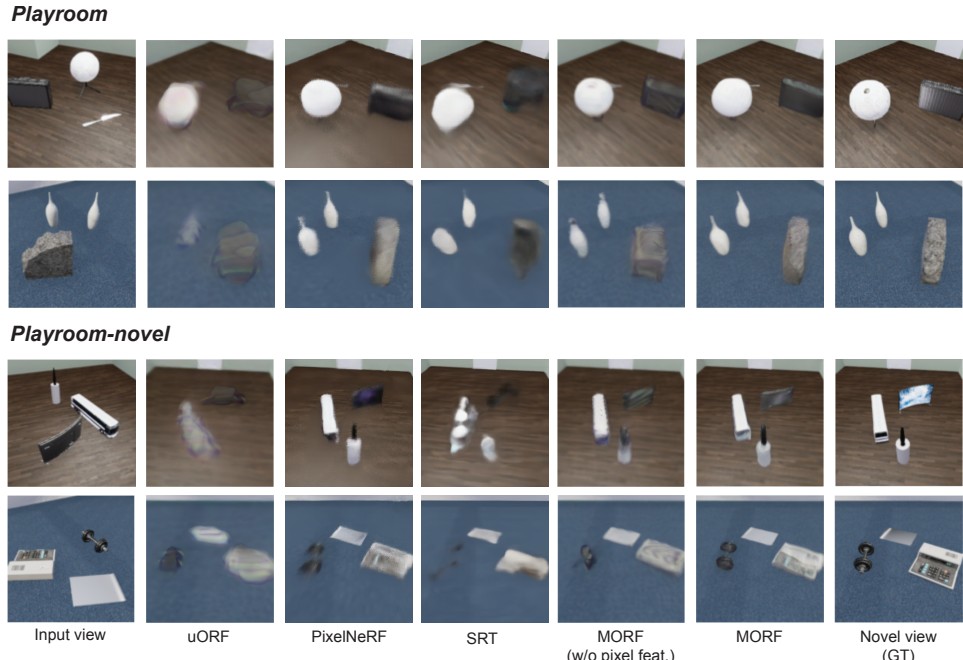

*Playroom-novel*

| Input view | uORF | PixelNeRF | SRT | MORF (w/o pixel feat.) | MORF | Novel view (GT) |

Figure 3: Qualitative results on novel view synthesis on the *Playroom* and *Playroom-novel* datasets. The objects in *Playroom-novel* datasets are not included in the training set. MORF outperforms the baseline models on both foreground and background reconstruction. MORF is better at reconstructing fine-grained textures and object geometries than the other models.

| Models | Playroom | | | | | Playroom-novel | | | | |
|---|---|---|---|---|---|---|---|---|---|---|
| | View synthesis | | | Geometry | | View synthesis | | | Geometry | |
| | LPIPS↓ | SSIM↑ | PSNR↑ | O-CD↓ | S-CD↓ | LPIPS↓ | SSIM↑ | PSNR↑ | O-CD↓ | S-CD↓ |
| uORF | 0.348 | 0.634 | 21.5 | 0.324 | 0.113 | 0.350 | 0.636 | 21.6 | 0.324 | 0.150 |
| pixelNeRF | 0.250 | 0.745 | 24.4 | - | 0.133 | 0.265 | 0.725 | **23.1** | - | 0.128 |
| SRT | 0.352 | 0.704 | 21.3 | - | - | 0.355 | 0.704 | 21.3 | - | - |
| MORF (no pix) | 0.244 | 0.735 | 23.0 | 0.239 | 0.096 | 0.264 | 0.722 | 22.2 | 0.260 | 0.122 |
| MORF (ours) | **0.161** | **0.784** | **24.5** | **0.208** | **0.078** | **0.189** | **0.755** | 22.9 | **0.224** | **0.110** |

Table 1: Quantitative comparison on novel view synthesis and geometry reconstruction. Novel view synthesis performance is measured by LPIPS, SSIM, and PSNR. Geometry reconstruction is measured by Object Chamfer Distance (O-CD) and Scene Chamfer Distance (S-CD). O-CD measures the geometry reconstruction quality of individual object meshes. S-CD measures the 3D layout of multiple objects in the scene by comparing the reconstructed and ground-truth scene meshes. For pixelNeRF (Yu et al., 2020), O-CD is not reported due to the lack of object decomposition.

**Metrics**  We compare the MORF and other baselines using Chamfer Distance (CD) (Sun et al., 2018). We compute two types of CDs, object mesh Chamfer Distance (O-CD) and scene mesh Chamfer Distance (S-CD). The O-CD metric focuses on measuring the quality of each individual mesh reconstruction, while the S-CD metric focuses on measuring the quality of the objects' layout in 3D. We compute S-CD on the foreground object meshes in a scene. Points are uniformly sampled on the mesh surface to create a dense point cloud; then $N$ points are randomly sampled from the point cloud, where $N$ is 1,024 per object mesh and 3,072 per scene mesh. We normalize the point cloud coordinates into a unit cube before CD calculation. Due to the lack of object decomposition, pixelNeRF only outputs a single mesh encompassing both the foreground objects and the background. For a fair comparison of S-CD with MORF, we remove the background meshes by setting the density of the grid points below a z-value threshold to zero before applying marching cubes. We search for the threshold with the best S-CD on a validation set, and use the threshold to calculate pixelNeRF's S-CD on the *Playroom* and *Playroom-novel* test datasets.

**Results**  We show the results in Table 1 and Figure 4. MORF outperforms all methods in terms of O-CD and S-CD. As seen in Figure 4, uORF (Yu et al., 2022) is only able to recover the coarse

*Playroom*

*Playroom-novel*

| Input view | uORF | PixelNeRF | MORF (w/o pixel feat.) | MORF | MORF (with GT masks) | GT |

Figure 4: Qualitative results on 3D segmentation and mesh reconstruction. MORF produces more accurate mesh reconstructions than uORF and pixelNeRF, as well as the ablated model of MORF trained without pixel features as conditioning. PixelNeRF only outputs a single mesh encompassing both the foreground objects and the background. We remove the background mesh in pixelNeRF and visualize the foreground objects only.

| Models | Move object | | | Replace object | | | Replace background | | |
|---|---|---|---|---|---|---|---|---|---|
| | LPIPS↓ | SSIM↑ | PSNR↑ | LPIPS↓ | SSIM↑ | PSNR↑ | LPIPS↓ | SSIM↑ | PSNR↑ |
| uORF | 0.381 | 0.573 | 20.7 | 0.384 | 0.575 | 20.8 | 0.371 | 0.593 | 20.4 |
| MORF (no pix) | 0.302 | 0.706 | 21.9 | 0.328 | 0.701 | 21.4 | 0.319 | 0.700 | 20.9 |
| MORF (ours) | **0.223** | **0.758** | **22.7** | **0.250** | **0.751** | **22.1** | **0.232** | **0.758** | **21.7** |

Table 2: Quantitative comparison on three scene editing tasks. MORF outperforms uORF on all the metrics.

geometry of the objects. pixelNeRF(Yu et al., 2020) tends to miss the fine details of small objects and thin objects, while MORF is able to learn more fine-grained object geometry given a single image.

### 4.3 SCENE EDITING

We consider three scene editing tasks: moving objects, replacing objects, and replacing the background. For moving and replacing objects, we follow the protocol from the uORF work (Yu et al., 2022) and select the object that has the largest IoU with the ground-truth masks of the target object for editing. pixelNeRF (Yu et al., 2020) is not comparable on scene editing due to the lack of object decomposition. We report LPIPS, SSIM and PSNR on the *Playroom-edit* dataset. Ground-truth masks are only used for selecting which object to edit. EISEN segmentation masks from a pretrained model are used in the feedforward pass of MORF on all the images. We show the results in Table 2 and Figure 5. MORF outperforms uORF on all metrics across the three editing tasks. MORF trained without pixel features results in blurry reconstruction on some of the objects.

### 4.4 ABLATION STUDIES

MORF performance on *Playroom* drops slightly when object latents are not used, though improves slightly on the O-CD metric. In contrast, pixel latents are crucial for MORF to perform well on both novel view synthesis and geometry reconstruction. This implies that although object-centricity provides a strong optimization constraint for learning object radiance fields, it is insufficient for modeling the fine details of object textures and shapes; local pixel features are critical for scaling to complex scenes.

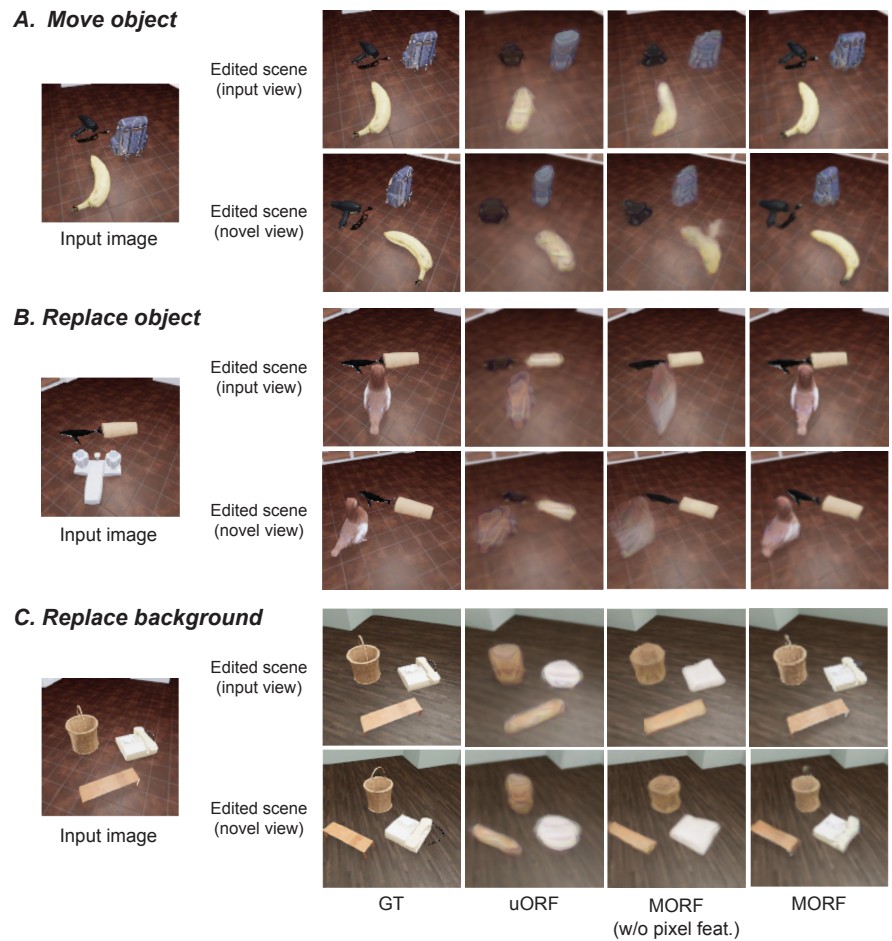

Figure 5: Qualitative results on three scene scene editing tasks. The synthesis of the edited scene from both input and novel views are shown in the top and bottom rows respectively. MORF is able to manipulate individual object radiance fields to generate novel scenes. Both uORF and MORF (without pixel features) are able to perform scene editing to a limited extent, but their reconstructions are blurry. This comparison shows the advantage of using both EISEN segmentations and pixel latents for scene editing tasks.

Finally, we compare MORF trained with EISEN segmentation masks to MORF trained with GT masks. Unsurprisingly, the latter shows better performance in all metrics. Qualitatively, we observe that MORF occasionally misses small objects or object parts, which is likely a direct consequence of EISEN failing to accurately segment these fine-scale scene elements. This indicates that improving unsupervised 2D segmentation should lead to further improvement of MORF's 3D representation learning.

| Mask types | slot feat. | pixel feat. | Playroom | | | | |
|---|---|---|---|---|---|---|---|
| | | | LPIPS ↓ | SSIM↑ | PSNR↑ | O-CD ↓ | S-CD ↓ |
| EISEN | ✓ | ✗ | 0.266 | 0.727 | 22.8 | 0.239 | 0.096 |
| EISEN | ✗ | ✓ | 0.170 | 0.776 | 24.3 | 0.198 | 0.079 |
| EISEN | ✓ | ✓ | 0.162 | 0.781 | 24.4 | 0.208 | 0.080 |
| GT | ✓ | ✓ | 0.140 | 0.791 | 25.0 | 0.147 | 0.036 |

Table 3: Ablation study results.

## 5 CONCLUSION

In this work, we propose the Movable Object Radiance Fields (MORF) model, which scales unsupervised 3D object-centric scene representation learning to complex and diverse multi-object scenes. We demonstrate that MORF enables faithful photometric and geometric reconstruction of scenes with unseen configurations from a single view, generalizes well to unseen object categories, and supports complex editing tasks. We believe our positive results suggest the promise of further scaling unsupervised 3D factorized representation learning to more complex scenes.

## REPRODUCIBILITY STATEMENT

To ensure the reproducibility of our work, we will release the training and testing code, as well as the data to reproduce our results upon publication.

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

# 6 SUPPLEMENTARY MATERIAL

## 6.1 PLAYROOM DATASET

The Playroom dataset is generated with ThreeDWorld (Gan et al., 2020) using custom code. The dataset contains 15,000 training scenes and 600 test scenes. Each scene includes 3 objects randomly sampled from a set of unique 2,086 object models in 231 object categories. The objects are rendered photo-realistically with complex geometry and textures. Table 4 and Figure 6 show the comparison between the Playroom dataset and the MultiShapeNet datasets.

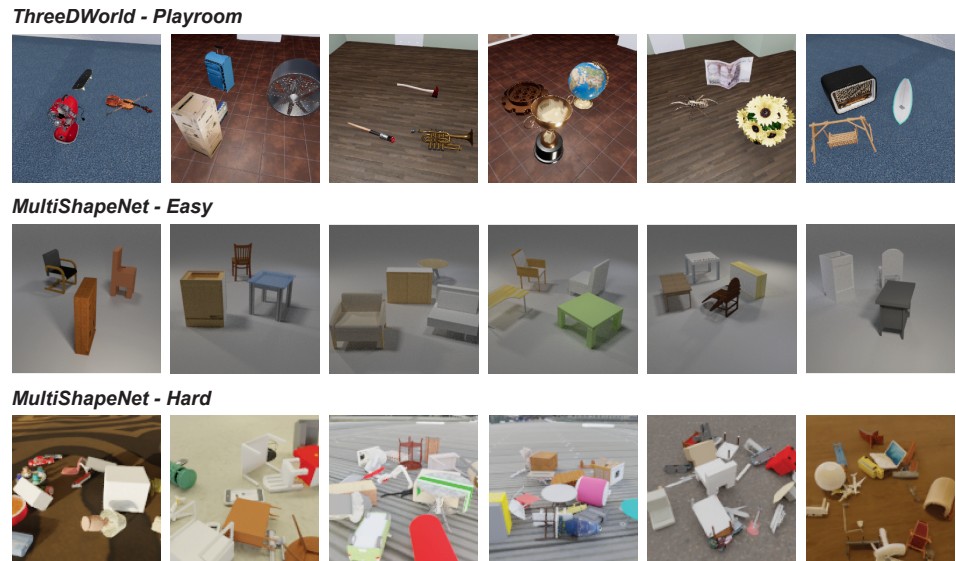

*ThreeDWorld - Playroom*

*MultiShapeNet - Easy*

*MultiShapeNet - Hard*

Figure 6: Qualitative comparison between Playroom, MultiShapeNet-Easy, and MultiShapeNet-Hard

We simulate 2 video frames for each scene, with a randomly selected object invisibly pushed at the first frame to generate object motion. At each time step, we render 4 views at resolution 128 with random camera azimuth angles. It is important to note that we

| Dataset | scenes | objects | categories | env. maps | obj. per scene | views |
|---------|--------|---------|------------|-----------|----------------|-------|
| Playroom | 15k | 2k | 231 | 3 | 3 | 4 |
| MSN-Easy | 80k | 12k | 3 | 1 | 2-4 | 3 |
| MSN-Hard | 1M | 51k | 55 | 382 | 16-32 | 10 |

Table 4: Dataset statistics comparison

only use the video frame pairs for training EISEN. We only use the static images at a single time step for learning radiance fields. Figure 7 shows examples of the video frame pairs from the dataset, along with the EISEN segmentations and ground truth segmentation masks.

## 6.2 STATIC SCENE SEGMENTATIONS IN 2D

We evaluate the segmentation quality of EISEN and compare it to other unsupervised segmentation methods: Slot attention (Locatello et al., 2020), unsupervised SAVi (Kipf et al., 2021), and DOM (Bao et al., 2022). To evaluate segmentation quality, we compute both the Adjusted Rand Index (ARI) and mean Intersection over Union (mIoU) metrics, following previous works in unsupervised 2D segmentations. We evaluate the metrics for both the foreground and background segmentation. As shown in Table 5 and Figure 9, Slot attention and unsupervised SAVi remain struggling with segmenting the Playroom objects. DOM uses optical flow to self-supervise the segmentation masks. However, it underperforms EISEN and outputs incomplete object segmentations, especially for objects with complex geometry details. The output segmentations of the baseline methods are not accurate enough to be used for training downstream object radiance fields.

## 6.3 STATIC SCENE SEGMENTATIONS IN 3D

We evaluate the quality of learned 3D scene representations using scene segmentations in 3D. The segmentations of uORF and MORF are obtained by first volume-rendering density maps of objects and background, and then assigning the pixel to the one with highest density. Table 5 shows the ARI and mIoU metrics. To reflect the 3D nature, we report both metrics on reconstructed input views and

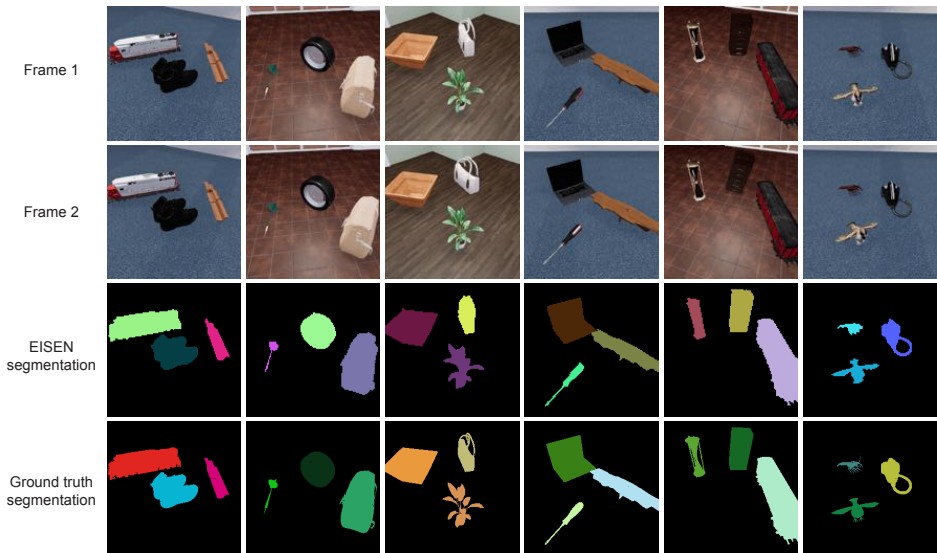

Figure 7: Playroom video frames, EISEN segmentations, and ground truth segmentations

| Models | Segments | Playroom | | | | Playroom-novel | | | |
|---|---|---|---|---|---|---|---|---|---|
| | | Input view | | Novel view | | Input view | | Novel view | |
| | | ARI↑ | mIoU ↑ | ARI ↑ | mIoU ↑ | ARI↑ | mIoU ↑ | ARI ↑ | mIoU ↑ |
| SlotAttn | 2D | 7.7 | 22.8 | - | - | 7.5 | 22.5 | - | - |
| Unsup. SAVi | 2D | 2.0 | 14.6 | - | - | 2.0 | 14.8 | - | - |
| DOM | 2D | 73.6 | 57.2 | - | - | 71.9 | 56.4 | - | - |
| EISEN | 2D | **93.3** | **87.3** | - | - | **92.2** | **87.4** | - | - |
| uORF | 3D | 76.2 | 67.1 | 64.1 | 59.1 | 75.8 | 66.9 | 63.4 | 58.4 |
| MORF (ours) | 3D | 91.5 | 84.6 | **79.1** | **72.7** | 90.7 | 84.6 | **71.8** | **66.4** |

Table 5: Quantitative comparison scene segmentations in 3D

novel views. MORF outputs more accurate segmentations than uORF. EISEN segmentations can only be compared in the input view. We observe EISEN is slightly better than MORF. MORF's 3D segmentations are slightly larger than the ground truth due to bilinear sampling in pixel conditioning.

### 6.4    GENERALIZATION TO REAL IMAGES

We go beyond the synthetic data to test the pretrained model's generalization on real images. Given a few real photos taken using a cellphone, we first use a unsupervised pretrained EISEN to compute the segmentations, followed by novel views reconstruction using pretrained MORF. We show the reconstructions and 3D segmentations in Figure 10. Our method is able to predict plausible geometry and segmentations for all the objects from a novel view.

### 6.5    ADDITIONAL EXPERIMENT ON MULTISHAPENET-EASY

In addition to the Playroom dataset, we train and test our model on the MultiShapeNet-Easy dataset (MSN-Easy) (Stelzner et al., 2021). uORF and pixelNeRF are also trained and evaluated for comparison. We train the models on 80,000 training scenes and report the novel view synthesis metrics on 500 validation scenes.

We cannot train EISEN on the MSN dataset since there is no motion. We thus train EISEN on the COCO dataset and use it directly on the MSN dataset. The segmentations for training are thus flawed; However, even with imperfect segmentation masks, we demonstrate that the reconstruction quality is significantly better than uORF and comparable to pixelNeRF in Figure 11.

| Model | LPIPS ↓ | SSIM↑ | PSNR↑ |
|---|---|---|---|
| uORF | 0.328 | 0.799 | 24.8 |
| pixelNeRF | 0.229 | **0.881** | **29.2** |
| MORF | **0.185** | 0.855 | 27.5 |

Table 6: Reconstruction metric

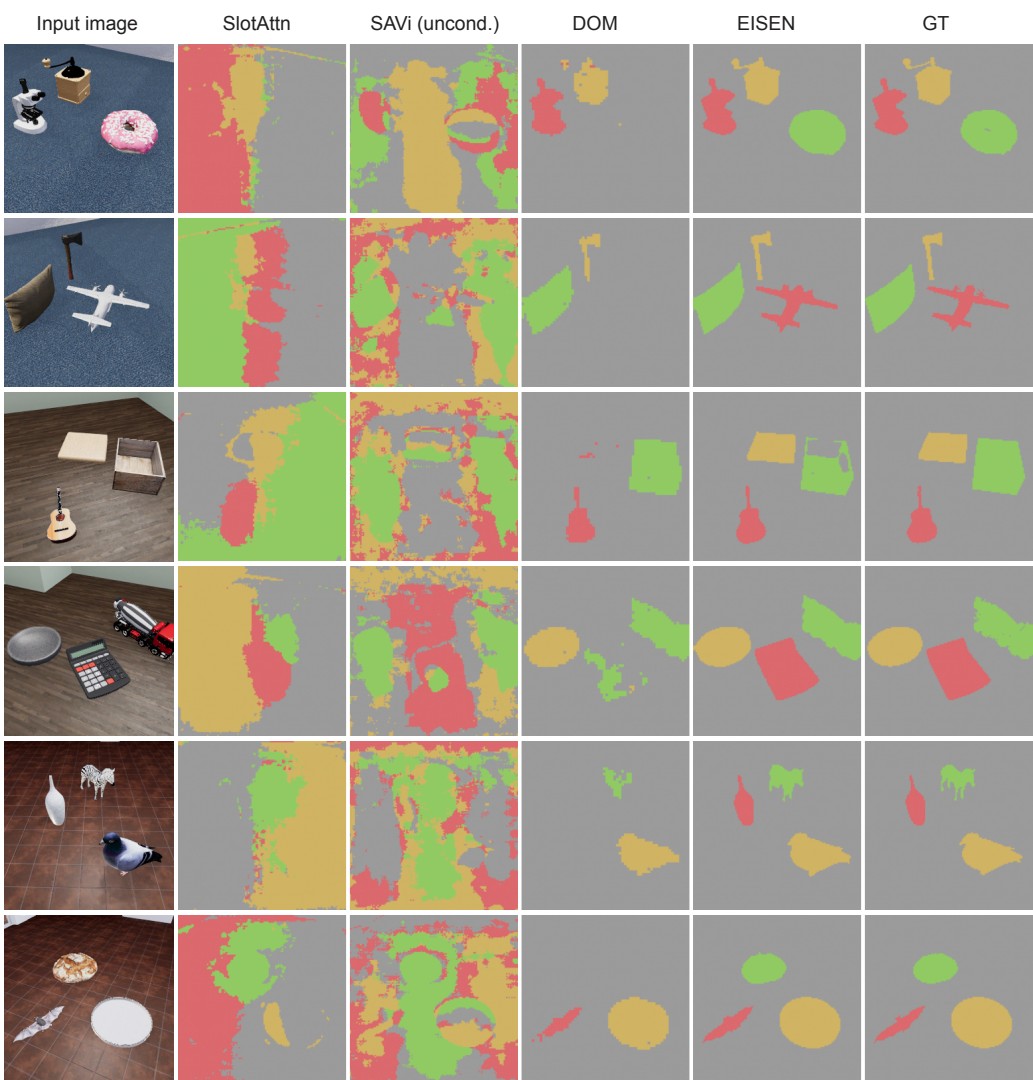

Figure 8: Qualitative results on static scene segmentation in 2D

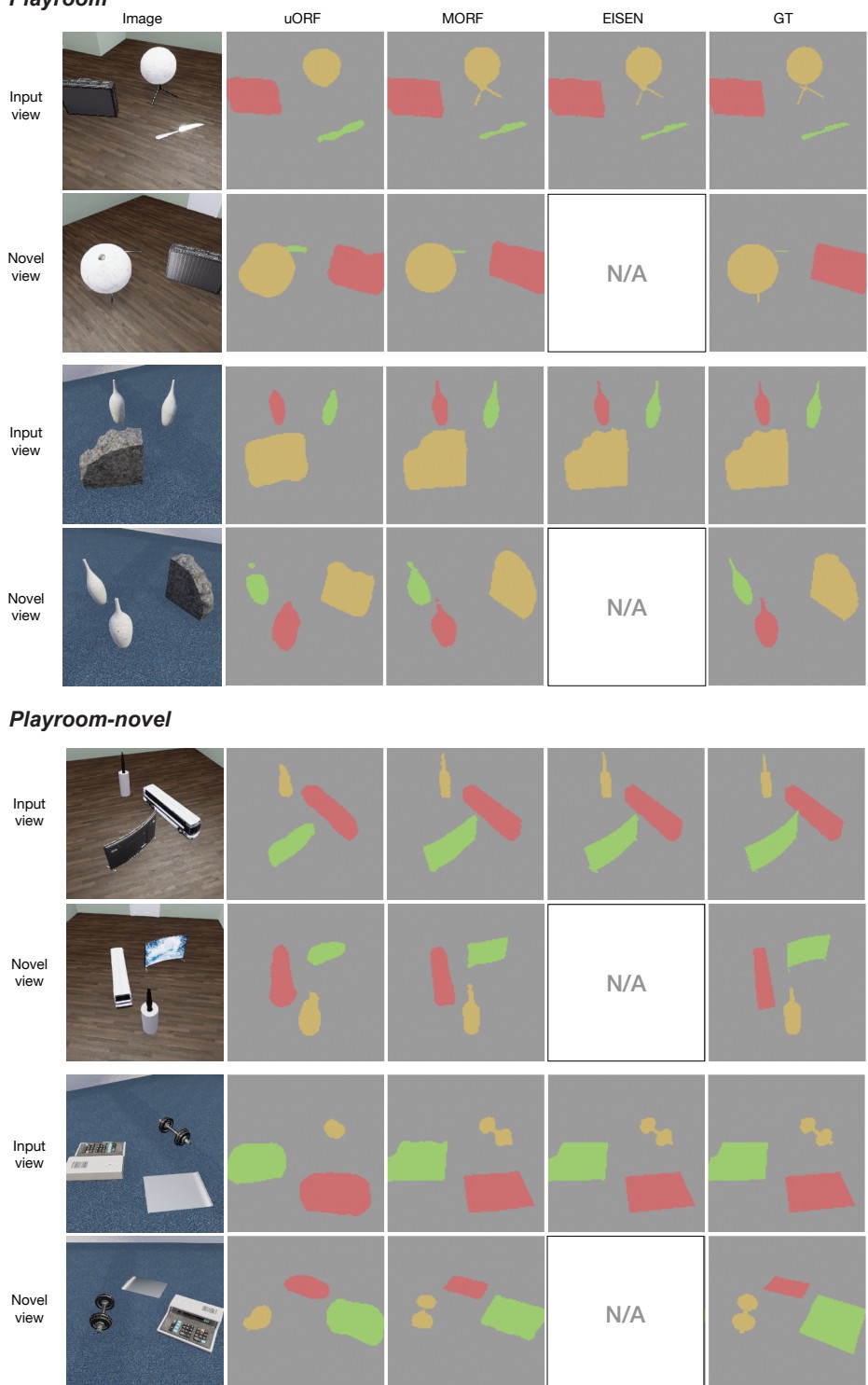

Figure 9: Qualitative results on static scene segmentation in 3D

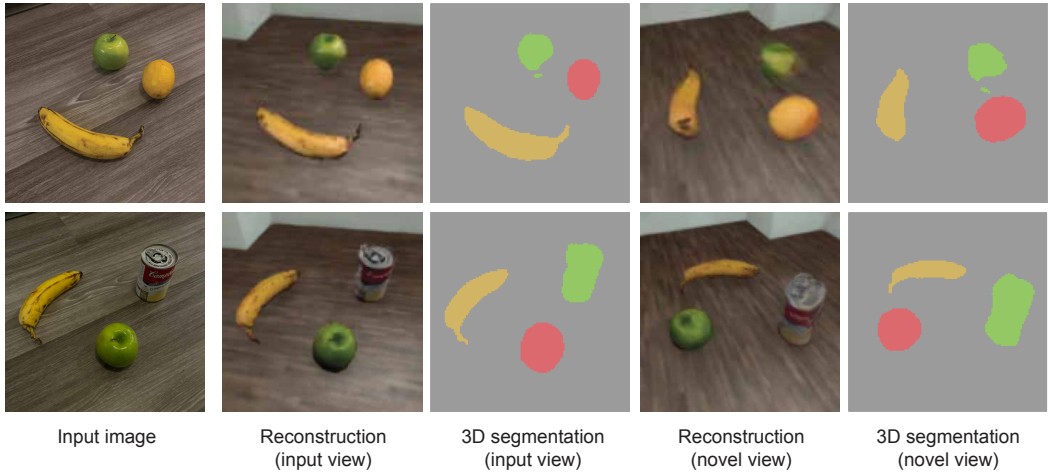

Figure 10: Novel view synthesis and 3D segmentations on real images taken using cellphone

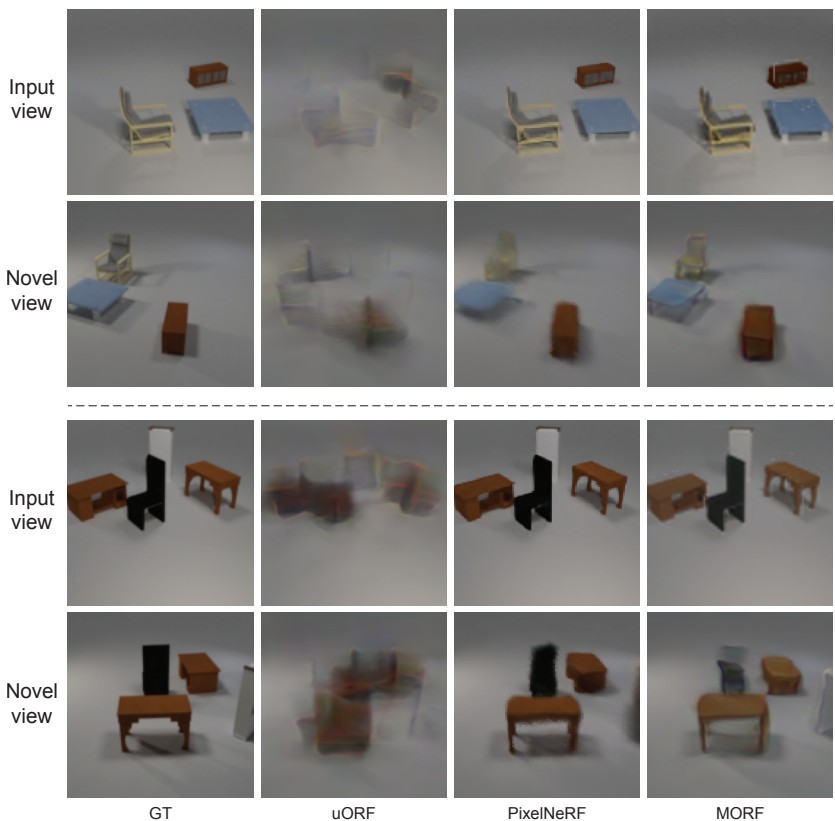

Figure 11: Novel view synthesis on MultiShapeNet-Easy

