# OpenReview forum: "Unsupervised 3D Scene Representation Learning via Movable Object Inference"
_ICLR.cc/2023/Conference — Submitted to ICLR 2023_

### Official Review · Reviewer_kstW · 2022-10-20

**Confidence:** 4
**Correctness:** 3
**Technical Novelty And Significance:** 2
**Empirical Novelty And Significance:** 2
**Recommendation:** 5

**Clarity, Quality, Novelty And Reproducibility:**

+ The paper is well-written in general. The reviewer is concerned with the novelty of the paper by considering it took a few components from existing works.

+ It would be great to explain the advantage of using the object level feature.

+ The method is reproducible.


**Details Of Ethics Concerns:**

I don't have ethics concerns about this paper

**Strength And Weaknesses:**

#Strength
+ The paper leverages the motion information for 3D object centric learning. It can handle images with complex textures

+ The object nerf and background nerf are trained based on global object feature and pixel-wise features.


#Weakness
+ Novelty of the paper. The proposed approach highly depends on the 2D segmentation mask which provides significant information for 3D object centric learning. Moreover, the 2D object masks are obtained from an existing work and the formulation for the object compositional nerf is not new [R1], which renders the proposed method lacks novelty. Please refer to the following missing reference for the compositional nerf [R1, R2].

 +  [R1] Unsupervised Discovery of Object Radiance Fields, ICLR2022
 + [R2] Learning Object-Compositional Neural Radiance Field for Editable Scene Rendering, ICCV2021.

+ The 2D mask segmentation framework highly depends on the quality of the optical flow. Then it cannot handle the images of pure color.

+ Eq. (4). Should ${\bf c}$ be ${\bf c}_i$?

+ It would be great to explain why results from the proposed approach cannot recover sharp boundaries and the reconstructed objects are larger than the real one (see Fig. 4)

+ It is hard to interpret the scene editing results. It looks like all views are different from the ones shown on the top? It makes it difficult for the reader to understand whether the images are rendered from changed objects’ positions or just the changed camera view point.


**Summary Of The Paper:**

This paper introduces a new framework for 3D object centric learning by leveraging the 2D object mask extracted from the motion information. The proposed approach then trains separate object and background nerf using different network structure. Experiments on data with more complex texture demonstrates the performance of the method.

**Summary Of The Review:**

This paper tackles a challenging problem, unsupervised 3D object-centric learning. The proposed framework is convincing. However, it largely depends on the 2D mask information obtained from a 2D image segmentation framework and the object compositional nerf is not new. Thus, the reviewer is concerned that there is not sufficient novelty in the paper.

---

> ### Author Response · Authors · 2022-11-19
> **Author response to Reviewer kstW**
>
> Thank you for constructive comments and feedback!  We have revised the paper with new experiments and discussions based on your feedback, and we respond to individual points below.
>
> > Novelty of the paper. The proposed approach highly depends on the 2D segmentation mask which provides significant information for 3D object centric learning. Moreover, the 2D object masks are obtained from an existing work and the formulation for the object compositional nerf is not new [R1], which renders the proposed method lacks novelty. Please refer to the following missing reference for the compositional nerf [R1, R2].
>
> We would like to respectfully clarify that we have cited the related work [R1, R2] in our original submission. [R1] is cited in the 4th line of the second paragraph in the introduction section. [R2] is cited in the 3rd line of the scene de-rendering paragraph in the related work section.
>
> The goal of this paper is to solve the important problem of unsupervised object-centric 3D representation learning from a single image of complex scenes, which remains a challenging and unsolved problem. Existing methods are not scalable to datasets with complex object textures, resulting in inaccurate 2D segmentations and 3D radiance field representations.
>
> The key innovation of our method is to decouple the 2D segmentation problem from learning 3D representations of objects. We show that once high-quality segmentations are obtained through an independent unsupervised method, we can learn 3D object-centric representations in complex category-agnostic scenes. This innovation addresses the limitation of the previous methods and allows substantial progress to be made on this difficult and important problem.
>
> > The 2D mask segmentation framework highly depends on the quality of the optical flow. Then it cannot handle the images of pure color.
>
> We clarify that optical flow is only needed during the training time of EISEN. No optical flow is required during the inference time. Once EISEN is trained, we only need a single RGB image (no optical flow) to compute the segmentation masks and to infer the radiance fields. Therefore, our method is not dependent on the quality of the optical flow and can handle images (and objects) of pure color. In Figure 3 of our main paper, we show that our method can deal with objects of pure color, such as the white sphere in the first row and the gray cardboard in the last row.
>
> > Eq. (4) should c be c_i?
>
> Thanks! We have fixed the typo in Equation 4.
>
> > Recovering sharp boundaries
>
> The reconstructed meshes are obtained by running the marching cube algorithm. The boundary sharpness is dependent on the spatial resolution for running the marching cube, which is 256 in our experiments. Please note that our geometry reconstruction is much better compared to baselines (Figure 4 in the main paper) even with limited spatial resolution of the marching cube.
>
> The reconstructed objects are larger than the real one (see Fig. 4).
> The meshes in Figure 4 are visualized in blender and rendered from a manually adjusted camera position. The reconstructed objects appear to be larger than the real ones since the camera is closer to the objects. We have adjusted the camera positions in blender and updated the figure accordingly (Figure 4).
>
> > In the scene editing results, it looks like all views are different from the ones shown on the top? It makes it difficult for the reader to understand whether the images are rendered from changed objects’ positions or just the changed camera view point.
>
> We update the scene editing figure (Fig. 5) and add caption for better clarity. The first image shows the input image of the scene. The other images show the edited scenes rendered from two different viewpoints: the input view and a randomly selected novel view respectively.
>
> > It would be great to explain the advantage of using the object level feature.
>
> As shown by the ablation studies in Table 3, adding object-level features slightly improves the quality of novel view synthesis. This is because object-level features allow the model to better handle occlusion and invisibility.
>
> Given a single input view for novel view synthesis. pixel-level features alone are insufficient to reconstruct novel views of invisible or occluded object parts in the input view, resulting in blurry reconstruction. On the other hand, object-level features provide global conditioning that allows the NeRF model to predict the density and color of objects even if parts of them are invisible or occluded.

---

### Official Review · Reviewer_NgLD · 2022-10-21

**Confidence:** 3
**Clarity, Quality, Novelty And Reproducibility:** The ideas are clear and reproducible,…
**Correctness:** 3
**Technical Novelty And Significance:** 2
**Empirical Novelty And Significance:** 2
**Recommendation:** 3

**Strength And Weaknesses:**

Strength:
This paper proposes a Movable Object Radiance Fields, which separate 3D scenes into objects and background before reconstruction. The authors claim that this method can better handle complex and diverse multi-object 3D scenes.


Weakness:
1.The overall structure takes advantage from many advanced algorithms, e.g.. object mask generation, object latent code computation, compositional rendering. I think the authors should better clarify the novelty and unique contributions of their method.

2.The authors carry out their experiments only on self-generated datasets. I think that for better comparison, they should test on more public datasets, e.g. the ShapeNet dataset, the DTU dataset, etc.

3.The authors use a long description to introduce the EISEN segmentation method. I think it would be better if they explain why EISEN, not other methods are chosen to for mask generation in the proposed method. Besides, they should provide the evaluation results for segmentation quality of EISEN; maybe even apply some more segmentation methods to show the influence of object masks (right now they only compare EISEN with GT masks without quantitative mask qualities).

**Summary Of The Paper:**

This paper proposes a Movable Object Radiance Fields, by using the EISEN method to generate object masks and processing object and background separately. Comparing with uORF and PixelNeRF methods on three self-generated datasets, the authors show that the proposed method can extract accurate object geometry and has higher performance in scene reconstruction and editing.

**Summary Of The Review:**

The proposed algorithm is not well compared and demonstrated.

---

> ### Author Response · Authors · 2022-11-19
> **Author response to Reviewer NgLD**
>
> Thank you for constructive comments and feedback!  We have revised the paper with new experiments and discussions based on your feedback, and we respond to individual points below.
>
> >The overall structure takes advantage from many advanced algorithms, e.g.. object mask generation, object latent code computation, compositional rendering. I think the authors should better clarify the novelty and unique contributions of their method.
>
> The goal of this paper is to solve the important problem of unsupervised object-centric 3D representation learning from a single image of complex scenes, which remains a challenging and unsolved problem. Existing methods are not scalable to datasets with complex object textures, resulting in inaccurate 2D segmentations and 3D radiance field representations.
>
> The key innovation of our method is to decouple the 2D segmentation problem from learning 3D representations of objects. We show that once high-quality segmentations are obtained through an independent unsupervised method, we can learn 3D object-centric representations in complex category-agnostic scenes. This innovation addresses the limitation of the previous methods and allows substantial progress to be made on this difficult and important problem.
>
> > The authors carry out their experiments only on self-generated datasets. I think that for better comparison, they should test on more public datasets, e.g. the ShapeNet dataset, the DTU dataset, etc.
>
> Thanks! Following your suggestion, we test our method on an independent dataset. We choose MultiShapeNet-Easy (MSN) since it is commonly evaluated by related works for object-centric 3D representation learning. MORF outperforms uORF and PixelNeRF (in LPIPS, uORF: 0.328, PixelNeRF: 0.229, MORF: 0.185, lower is better). Note that MSN does not have groundtruth mesh so that we cannot evaluate geometry where our method significantly advances over existing methods. Also note that we cannot train EISEN on the MSN dataset since there is no motion. We thus train EISEN on the COCO dataset and use it directly on MSN dataset. The segmentations for training are thus flawed; However, even with imperfect segmentation masks, we demonstrate that the reconstruction quality is better than uORF in Appendix 6.5, Figure 11.
>
> > The authors use a long description to introduce the EISEN segmentation method. I think it would be better if they explain why EISEN, not other methods are chosen to for mask generation in the proposed method. Besides, they should provide the evaluation results for segmentation quality of EISEN; maybe even apply some more segmentation methods to show the influence of object masks (right now they only compare EISEN with GT masks without quantitative mask qualities).
>
> EISEN is chosen for mask generation since it’s the state-of-the-art method for learning high-quality object segmentations without ground-truth supervision. Existing unsupervised methods do not scale well to complex datasets. In addition, supervised models generalize poorly to novel scenes and categories. The segmentation quality of these methods is not suitable for learning downstream 3D representations.
>
> Following your suggestion, we evaluate the segmentation quality of EISEN and compare it to other unsupervised segmentation methods: Slot attention [1], unsupervised SAVi [2], and DOM [3]. Slot attention and unsupervised SAVi struggle with segmenting the Playroom objects. DOM requires optical flow, yet it generates incomplete segmentations especially for objects with complex geometry details. We add the quantitative ARI and mIoU metrics in Appendix 6.2, Table 5, and the segmentations in Figure 8. EISEN performs the best, and the output segmentations of the baseline methods are not accurate enough for training downstream object radiance fields.  This further illustrates the novelty of our proposed method.
>
> [1] Object-Centric Learning with Slot Attention, NeurIPS 2020
>
> [2] Conditional Object-Centric Learning from Video, ICLR 2022
>
> [3] Discovering Objects That Can Move, CVPR 2022

---

### Official Review · Reviewer_dyX9 · 2022-10-25

**Confidence:** 3
**Correctness:** 4
**Technical Novelty And Significance:** 2
**Empirical Novelty And Significance:** 3
**Recommendation:** 6

**Clarity, Quality, Novelty And Reproducibility:**

The paper is clearly written, is high quality, and the authors state that they will release code and data on acceptance, so reproducibility  should not be a problem.

This work does seem to be a fairly straightforward combination of  EISEN (Chen et al, 2022) for detecting 2d masks of arbitrary objects and uORF (Yu et al, 2022) for object-based scene reconstruction.  Their Loss function is almost identical to Yu with the addition of the adversarial loss component.  There is no ablation to indicate how important that additional loss is so it's hard to know how significant that addition is.  I would love to see this added in future versions.

The similarity to the work of Yu is not called out very clearly. They mention Yu in the related work section and is one of the primary baseline methods used, but during the discussion of the method implementation there is little indication of the similarity to Yu, and it is left to the reader to put this together.



**Strength And Weaknesses:**

The strengths of this paper are as follows:
* Allows for scene rendering from a single image and scene editing.
* Enables a new level of capability by generalizing previous work to handle multiple and arbitrary classes of objects in a scene rather than specific objects trained in a supervised fashion.
* Convincing quantitative and qualitative results demonstrating that the approach works as expected.


The weaknesses of this approach are as follows:
* All results are demonstrated only on synthetic data.

**Summary Of The Paper:**

This paper extends previous work on 3d scene decomposition and reconstruction from a single rgb image. Being able to reconstruct the 3d structure of a scene from a single rgb enables a host of new functionality such as re-rendering multiple views of the scene, editing the scene, and potentially generating images for training 2d and 3d inference models. This work is novel in that it generalizes previous work to arbitrary "movable" objects rather than relying on a supervised model trained on a fixed set of objects.  This is a step function improvement in capability enabled by this work.

**Summary Of The Review:**

This is a straightforward combination of two techniques that enables a step-function capability in 3d scene reconstruction from rgb images. I think this is a solid contribution and worth accepting at ICLR.

---

> ### Author Response · Authors · 2022-11-19
> **Author response to Reviewer dyX9**
>
> Thank you for constructive comments and feedback!  We have revised the paper with new experiments and discussions based on your feedback, and we respond to individual points below.
>
> > All results are demonstrated only on synthetic data.
>
> We go beyond the synthetic data to test the pretrained model’s generalization on real images. Given a real photo taken using a cellphone, we use our pretrained MORF model for inference. We show reconstructions and 3D segmentations of a few real photos in Appendix 6.4 Figure 10. Our method is able to predict plausible geometry and segmentations for all the objects from a novel view.
>
> > Their Loss function is almost identical to Yu with the addition of the adversarial loss component. There is no ablation to indicate how important that additional loss is so it's hard to know how significant that addition is. I would love to see this added in future versions.
>
> We clarify that our loss function does not include the adversarial loss component. Yu’s work includes the adversarial loss component. We find empirically that adding an adversarial loss component doesn’t lead to significant improvements, and it makes the convergence of the training slower. Therefore, we chose not to include the adversarial loss.
>
> > The similarity to the work of Yu. They mention Yu in the related work section and is one of the primary baseline methods used, but during the discussion of the method implementation there is little indication of the similarity to Yu, and it is left to the reader to put this together.
> We agree on the importance of clarifying the differences.
>
> Our approach is significantly different from Yu in decoupling the 2D segmentation problem from 3D radiance field learning. We have revised our paper in the method section to incorporate the following comparison:
>
> Differences:
>
> 1. Yu’s approach learns object decomposition and radiance fields in an end-to-end manner using the reconstruction loss. Our approach disentangles object decomposition from RGB reconstruction. We use high-quality unsupervised segmentation as an optimization constraint for radiance field learning.
>
> 2. As a result of (1), Yu’s approach is limited to using slot-based features for conditioning the object-centric NeRF models, resulting in blurry reconstructions of complex textures. With segmentation masks from EISEN, our method is able to compute pixel-level features from masked images as additional local conditioning, allowing accurate reconstruction of fine-grained textures.
>
> Similarity
> 1. Following Yu’s approach, we use two separate conditional NeRFs for reconstructing background and foreground objects respectively, which was found to improve reconstruction quality.
>
> 2. We adopt the same compositional rendering technique from Yu to combine the background and foreground object radiance fields.

---

### Official Review · Reviewer_Kin7 · 2022-10-25

**Confidence:** 4
**Correctness:** 3
**Technical Novelty And Significance:** 3
**Empirical Novelty And Significance:** 3
**Recommendation:** 6

**Clarity, Quality, Novelty And Reproducibility:**

__Clarity__: In general, the paper is well written and easy to follow. Some additional data would be useful to provide to make it easier to evaluate qualitative results:
 - Dataset characteristics such as resolution, number of video frames, etc.
 - Training time / computational cost of EISEN and MORF.
 - Videos from the dataset, as well as EISEN segmentation results compared with ground truth segmentations.
 - Videos showing novel view synthesis and segmentation results.
 - Extracted 3D meshes.

__Quality__: The experiments are setup in a reasonable way, although the inclusion of an SRT baseline as discussed above would be nice. I think not comparing to ObSuRF or OSRT is acceptable, as they operate in slightly different settings. Even though this is largely provided by EISEN, I think a segmentation comparison between uORF, EISEN, and MORF would still be useful, as the geometric scores blend together segmentation and 3D reconstruction quality.

__Novelty__: The fact that MORF mainly leverages cues from an existing, pretrained model reduces its novelty. That said, it is a first step towards the challenging setting of 3D video data.

__Reproducibility__: The authors have promised to provide code and data, which goes a long way for reproducibility. Neither has been provided for review, however. Taking the paper by itself, many implementation details are missing, such as the architectures of the image encoder, or the NeRF MLPs.


**Strength And Weaknesses:**

Strengths:
 - The paper addresses a relevant and interesting problem, namely, unsupervised object-centric 3D scene understanding.
 - It is well written and easy to follow.
 - The idea of incorporating motion cues for better segmentations makes sense and has not been explored for 3D models so far.
 - MORF convincingly outperforms uORF on the new dataset.

Weaknesses:
 - The characterization of prior datasets ("They only demonstrate simplistic scenes with a single object category, and the objects in the scene are in uniform colors.") is not accurate. While they are synthetic and might indeed by called simplistic, e.g. Multishapenet from ObSuRF and MSN-hard from OSRT use objects from multiple ShapeNet categories, and their standard, multi-colored textures.
 - Consequently, I disagree that the new dataset represents the jump in complexity that the paper suggests it does. To me, it appears of a similar level of complexity as ObSuRF's Multishapenet, and much simpler than OSRT's MSN-hard.
 - Increased performance comes at the cost of requiring video data for training, which the baseline models don't.
 - The fact that the processing of video data is outsourced to an existing 2D model decreases MORF's technical novelty, and doesn't make use of the full potential of combining 3D with video observations.
 - Given the emphasis on novel view synthesis in the evaluation, the state of the art model SRT (Sajjadi et al., 2021) should be included as a baseline, even though it will not yield segmentations or 3D geometry in its default configuration.
 - Some additional information could be added for clarity (see below).

**Summary Of The Paper:**

The paper proposes a new model for unsupervised object-centric 3D scene understanding, called MORF. Unlike previous methods, it leverages 2D image segmentation masks from the pretrained optical flow method EISEN. These are used to initialized a slot based representation augmented by pixel features (as in PixelNeRF), which are used to condition NeRF decoders for objects and backgrounds. On a novel synthetic 3D dataset called Playroom, the model yields better reconstructions and 3D geometry results when compared to prior work. Improved scene editing capabilities are demonstrated, as well.

**Summary Of The Review:**

Overall, the paper represents an interesting first attempt at utilizing both video and 3D cues for better unsupervised scene understanding. While I believe its novelty is somewhat overstated, the experiments are competently executed. I am therefore leaning towards acceptance.

---

> ### Author Response · Authors · 2022-11-19
> **Author response to Reviewer Kin7**
>
> Thank you for constructive feedback!  We respond to individual points below.
>
> > The characterization of prior datasets is not accurate
>
> Thank you for raising this point. We revised the text to correct the claim. Existing methods are only demonstrated to work well on simplistic scenes. However, it remains challenging to obtain accurate reconstructions on more complex datasets such as MultiShapeNet-Easy, and MultiShapeNet-Hard is even more challenging.
>
> > The complexity of the new dataset
>
> The Playroom dataset and the MultiShapeNet are complex in different ways. MultiShapeNet-Easy contains ~12k unique shapes from only three categories. The Playroom dataset contains ~2k unique objects from 231 unique categories. MSN-Hard contains ~51k unique models from 55 categories. We agree that MSN-hard is more difficult in terms of the number of unique object models and scene complexity.  Please refer to Table 4 and Figure 6 of Appendix 6.1 for a comparison between the Playroom and the MultiShapeNet datasets.
>
> > Increased performance comes at the cost of requiring video data for training
>
> We acknowledge the baseline models don’t require video data for training. However, please note that our method only uses pairs of video frames for training EISEN. Once EISEN is trained, we only need a single static RGB image to compute the segmentation masks and the radiance fields.
>
> > Novelty: the processing of video data is outsourced to an existing 2D model
>
> We clarify that the goal of this paper is to solve the important problem of unsupervised object-centric 3D representation learning from a single image of complex scenes, which remains an unsolved problem. Existing methods are not scalable to datasets with complex object textures, resulting in inaccurate 2D segmentations and 3D radiance field representations.
>
> The key innovation of our method is to decouple the 2D segmentation problem from learning 3D representations of objects. We show that once high-quality segmentations are obtained through an independent unsupervised method, we can learn 3D object-centric representations in complex category-agnostic scenes. This innovation addresses the limitation of the previous methods and allows substantial progress to be made on this difficult and important problem.
>
> Our method is conceptually novel in decoupling 2D segmentation from 3D representation learning. We demonstrate that using video in learning the 2D segmentations alone, but neither in 2D inference nor 3D learning, has already led to promising improvement in the learned 3D representations.
>
> We fully agree that combining 3D with video observations in a more integrated way is an exciting and promising future direction, and would love to explore it in the future.
>
> > state of the art model SRT
>
> Thanks!  As suggested, we train and evaluate the improved SRT model as discussed in Appendix 4 of the OSRT paper. We follow the training procedures for training SRT on the Playroom dataset, with the difference of training for 1M steps instead of 4M steps. We observed that the reconstruction metrics plateau after 0.5M steps. The comparison between SRT and our method will not be affected by prolonged training. SRT underperforms our method in terms of reconstruction metrics. We include the SRT results in Figure 3 and discussions in the experiment section.
>
> > Some additional data would be useful to provide
>
> * Dataset characteristics
>
> The Playroom dataset has a resolution of 128x128.  It contains 15000 training scenes. Each scene consists of 2 video frames, with a randomly selected object invisibly pushed at the first frame to generate motion. At each time step, we render 4 views with random camera azimuth angles.
>
> * Training time / computational cost of EISEN and MORF.
>
> Training EISEN on the Playroom dataset takes 20 hours on 1 A40 GPUs. Training MORF takes 7 days on 8 A40 GPUs.
>
> * Videos from the dataset, as well as EISEN segmentation results compared with ground truth segmentations.
>
> We show the frame pairs from the dataset, EISEN and ground truth segmentations in Appendix 6.1, Figure 7.
>
> * Videos showing novel view synthesis and segmentation results.
>
> We upload a video in the supplementary material showing novel view synthesis and segmentation results.
>
> * Extracted 3D meshes.
>
> The extracted 3D meshes of MORF and other baseline methods are visualized in Figure 4.
>
> > segmentation comparison between uORF, EISEN, and MORF
>
> We have included the segmentation metrics of uORF, EISEN and MORF in Appendix 6.3, Table 5. The segmentations of uORF and MORF are obtained by first volume-rendering density maps of objects and background, and then assigning the pixel to the one with highest density. MORF outputs more accurate segmentations than uORF. EISEN segmentations can only be compared in the input view. We observe that EISEN is slightly better than MORF.  MORF’s 3D segmentations are slightly larger than the ground-truth due to bilinear sampling in pixel conditioning.

---

### Author Response · Authors · 2022-11-19
**Summary of Revision**

We thank the reviewers again for their constructive comments and feedback. We are eager to incorporate their suggestions as revision to our paper (highlighted in blue in the updated submission). We summarize the revisions below:

**Experiments:**
1. [NgLD, kstW, kin7]: In Appendix 6.2 and 6.3, we evaluate and visualize the 2D and 3D segmentations of our model and the baselines.
2. [dyX9]: In Appendix 6.4, we demonstrate MORF’s ability to generalize to real-world images.
3. [NgLD]: In Appendix 6.5, we evaluate MORF and baseline models on the MultiShapeNet-Easy dataset.
4. [Kin7]: We add the results and discussion for the SRT baseline in the experiment section.

**Writing:**
1. [kstW, dyX9]: In the method section, we fix the typo in Equation 4 and indicate the relationship to Yu’s approach
2. [kstw]: We update Fig. 4 and 5 for better clarity
3. [Kin7]: In Appendix 6.1, we include more information about the Playroom dataset and its comparison to other datasets.

**Supplementary material**
1. We add a video showing the novel view synthesis and segmentation results by rotating camera around the center of the scene.

We respond to suggestions and questions from individual reviewers below. We appreciate your time and effort in helping us improve the paper.

---

### Decision · Program_Chairs · 2023-01-20

**Decision:**

Reject

**Justification For Why Not Higher Score:**

Not enough novelty.

**Justification For Why Not Lower Score:**

N/A

**Metareview: Summary, Strengths And Weaknesses:**

The paper proposes a new model for unsupervised object-centric 3D scene understanding, called MORF. Unlike previous methods, it leverages 2D image segmentation masks from the pretrained optical flow method EISEN. These are used to initialized a slot based representation augmented by pixel features (as in PixelNeRF), which are used to condition NeRF decoders for objects and backgrounds. On a novel synthetic 3D dataset called Playroom, the model yields better reconstructions and 3D geometry results when compared to prior work. Improved scene editing capabilities are demonstrated, as well.

The paper addresses a relevant and interesting problem. It is well-written and easy to follow. The idea of incorporating motion cues for better segmentations makes sense and has not been explored for 3D models so far. MORF convincingly outperforms uORF on the new dataset.

The characterization of prior datasets is not accurate. The new dataset represents the jump in complexity that the paper suggests it does. Other state-of-the-art models should be included as a baseline. More information could be added for clarity. All results are demonstrated only on synthetic data. Clarify what novelty and contributions come from this article and not prior work. Use public datasets. The novelty of the paper. It is hard to interpret the scene editing results.

The authors have done a good job answering to the reviewers. However, these still are not fully convinced about the novelty.